# Effects of user behaviors on accumulation of social capital in an online social network

**Yuri Rykov**[1]*, **Olessia Koltsova**[2], **Yadviga Sinyavskaya**[2]

**1** The Centre for Population Health Sciences, Lee Kong Chian School of Medicine, Nanyang Technological University, Singapore, Singapore, **2** The Social and Cognitive Informatics Laboratory, National Research University Higher School of Economics, Saint Petersburg, Russia

* yuri.rykov@ntu.edu.sg, rykyur@gmail.com

**Data Availability Statement:** The data used in this study are available from the Open Science Framework: https://osf.io/hw2b6/ DOI 10.17605/OSF.IO/HW2B6.

## Abstract

The use of social network sites helps people to make and maintain social ties accumulating social capital, which is increasingly important for individual success. There is a wide variation in the amount and structure of online ties, and to some extent this variation is contingent on specific online user behaviors which are to date under-researched. In this work, we examine an entire city-bounded friendship network (N = 194,601) extracted from VK social network site to explore how specific online user behaviors are related to structural social capital in a network of geographically proximate ties. Social network analysis was used to evaluate individual social capital as a network asset, and multiple regression analysis–to determine and estimate the effects of online user behaviors on social capital. The analysis reveals that the graph is both clustered and highly centralized which suggests the presence of a hierarchical structure: a set of sub-communities united by city-level hubs. Against this background, membership in more online groups is positively associated with user's brokerage in the location-bounded network. Additionally, the share of local friends, the number of received likes and the duration of SNS use are associated with social capital indicators. This contributes to the literature on the formation of online social capital, examined at the level of a large and geographically localized population.

## Introduction

### Background

Social network sites (SNSs) are playing an important role in gaining and maintaining interpersonal relationships and obtaining related social outcomes. One such outcome is social capital, broadly understood as access to and use of social ties, which facilitate achievement of specific goals and acquisition of benefits [1–4]. The existing research has shown that social capital relates to a wide range of positive implications, such as individual health and longevity [5, 6], economic wealth [7] and educational achievements [8]. Likewise, online social capital–a fraction of social capital that is gained and/or maintained in an online social network–is also related to positive offline patterns. Thus, Facebook users with larger and denser friendship ego-networks tend to have a higher socioeconomic status [9], while a lower mortality rate is observed among users receiving more friendship requests [10].

**Funding:** This research was supported by the Basic Research Program at the National Research University Higher School of Economics (https://brp.hse.ru/) in 2017, the collective grant 3-68 "Internet as a socio-technical phenomenon". The funders had no role in study design, data collection and analysis, decision to publish, or preparation of the manuscript.

**Competing interests:** The authors have declared that no competing interests exist.

The amount of online social capital is, in turn, associated with online activities, including the general intensity of SNS use [11–13]. Although this does not increase the size of personal social networks beyond a certain limit (the Dunbar's number) [14–16], there are still significant differences in the number and composition of online social ties among users, and to some extent these differences depend on specific online user behaviors. Recent research [12, 17–22] has employed a variety of fine-grained metrics and identified specific user practices that have varying effects on social capital. However, there are still gaps in the knowledge regarding mechanisms connecting SNS user practices to social capital. The aim of this study is to determine what types of user behaviors and SNS use contribute most to users' online social capital.

We employ one of the most empirically grounded approaches to define social capital, viewing it as a structural asset, or as an advantage in terms of benefits that directly result from the structure and composition of one's social network. This definition is opposed to the vision of social capital as an amount of resources, usually self-reported by an individual (perceived social capital), and based on multiple studies showing the link between structures and benefits [3, 4, 23]. Different structural configurations of social ties facilitate different benefits. Burt argues that individual advantage is created by the way in which people are connected and identifies two structural sources of social capital: *network closure* and *brokerage*. Closure is a network's feature of being a bounded and tightly connected group of individuals. Closure facilitates better cooperation, resource mobilization and trust, because these forms of behavior are stimulated by the threat of sanctions among people with many common friends. Brokerage is a network position that bridges otherwise segregated and heterogeneous groups. Brokerage capacity–the amount of non-redundant contacts accessed and bridged by an actor–depends on the number of *structural holes* around an individual, which are gaps between disconnected parts of a broader network [4]. Brokerage capacity reflects the diversity of accessible social contexts, opinions, activities and resources. Unlike Burt, Lin argues that individual social capital should be determined rather by the entire network macro-structure of the population and by the individual's position within it rather than by the micro-structure of an individual's immediate environment. This is because valuable social resources are distributed unevenly within the entire population and might be accessed by indirect social connections [3]. It therefore makes sense to measure individual social capital both as local and global centralities in a large, but meaningful network.

As an example of such an online network, we choose a population bounded by a city, which is both large and meaningful. First, this approach reflects the general embeddedness of social capital in geographically proximate environments, such as a neighborhood, village or city, which is demonstrated in a large number of works [3, 11, 24–28]. Other studies have shown that online friendship and interaction, despite the potentially global character of SNSs, also tend to be geographically proximate [29–30]. Second, the city-level approach allows accounting for the aforementioned effect of indirect ties in a macro societal context–ties that provide knowledge of someone who knows the "right" person [24]. Simultaneously, it allows us to limit all online ties that are of low cost to establish and therefore occasionally very weak, by geographically proximate relationships that are more likely to provide access to tangible resources and location-related aid [24, 28] including finding jobs [31], available housing rentals, medical services [32] or childcare opportunities [33]. Of course, the extent to which the data derived from SNSs, telecommunications companies and from other digital traces represent human social networks as a whole, it is still a matter for investigation [34–36]. However, as SNSs are now an integral part of everyday life, social capital accumulated through them deserves research per se, even if it happens to be distinct from its offline counterpart.

## Hypotheses: Online user behaviors and network social capital

Summarizing the results of studies over the past two decades, Liu et al. [37] conclude that both social information seeking (i.e. browsing profiles of those individuals whom the user knows something about from an offline context in order to learn more about them) and social information disclosing have a positive effect on perceived online social capital. Among many types of social information, identity information (such as hometown, place of education, key biography events or user interests) is the one that may provide missing social context cues and facilitate establishing common ground and further tie formation between the parties, thus serving as social lubricant. For instance, Lampe et al. [38] showed that filling profile fields on Facebook was positively associated with the number of Facebook friends. Ellison et al. [11] treat such signaling to others about particular interests, affiliation to some organization or mutual social connections as a way to establish the connection with Friends of Friends, thereby transforming so-called "latent ties" (social ties that are "technically possible but not activated socially") [39, p. 137] into more "salient"—weak or strong- ties.

The disclosure of personal information online might be constrained by the privacy attitudes of SNS users [40]. The context collapse (i.e. the co-presence of different social groups in the shared online environment) may provoke those who are concerned with privacy issues to censor their disclosures [41] or limit the access to them by applying the advanced privacy settings [42]. Having the "friends-only" account or limiting the access to personal information or updates to specific groups of online friends turned out to be beneficial in terms of bonding social capital only for those who collected a high proportion of «actual» friends in their network [43]. At the same time, the distribution of content, only for specific groups of friends (i.e. applying the segmented privacy settings), leads to lower perceptions of bridging social capital [43].

Thus, in terms of social capital metrics, we expect that public self-disclosure of identity information may facilitate both network closure and brokerage–through joining well-connected homogeneous groups (e.g. classmates) or connecting to non-redundant contacts (e.g. people with rare interests). This lets us formulate our first hypothesis:

**H1:** The amount of publicly available identity information in a user's profile is positively related to his/her social capital.

Research on the role of *specific communication features* for social capital has shown mixed results. For instance, Burke et al. [17] investigated the effects of three distinct types of SNS use: *directed communication* which consists of personal, one-on-one exchanges (messages, likes etc.), *broadcasting* (information sharing with a broad audience) and *passive consumption* of social news. The authors found that only the amount of *incoming directed communication* acts had an impact on bridging social capital, i.e. ties connecting separated groups and fostering getting new information.

Other authors have mostly been studying outgoing communication and demonstrating its importance for social capital in a number of (contradictory) aspects. Thus, Lee et al [44] showed that bonding capital (belonginess to a tightly connected group) was higher among those who used the *Like feature* more frequently and *Comment feature* less frequently, while bridging capital was associated with *posting on a friend's wall*. However, Su and Chan [21] have demonstrated that *commenting*, along with *liking* and *sharing* were positively related to both bonding and bridging social capitals. Bohn et al. [45] found that the number of communication partners was positively associated with both network brokerage and closure in the interaction network, but the number of personalized outgoing communication ties had a positive effect only on brokerage. Apart from this, *Facebook relationship maintenance behavior* (FRMB), defined as a form of social grooming–an attention-signaling activity and engagement

with a user's friend network through direct communication (such as likes, comments or posts on a friend's wall), was found to be positively and strongly related to both bridging and bonding social capital [18–20, 46]. Outgoing communication has received more attention than incoming communication. We assume, that the effect of outgoing communication (broadcasting) on a user's social capital is contingent, i.e. more intense broadcasting will lead to higher brokerage if it reaches and attracts external audience, and–to higher network closure if it concerns more a user's existing friends. Meanwhile, contributions of others to a user's wall–a part of incoming communication available for research–may be expected to have a twofold effect on the wall owner's social capital. First, by getting acquainted and befriending each other, wall visitors may contribute to the wall owner's network closure [19]. Second, if the wall content and especially contributions to the wall are not restricted by a user to an already existing friend network, a vivid wall activity may attract newcomers who, after initial communication, may send or receive friend requests to/from the wall owner. This leads us to our second hypothesis on the role of communication activity comprised of two sub-hypotheses for outgoing (broadcasting) and incoming communication:

**H2a:** The amount of outgoing communication (broadcasting) is positively related to a user's social capital.

**H2b:** The amount of incoming communication (engagement of others in communication on a user's wall) is positively related to a user's social capital.

Although online group membership, as an SNS feature, should theoretically be important for social capital [47, 48] it has been receiving a modest amount of attention from researchers. Some studies suggest that participation in online groups should somehow facilitate networking behavior, because the groups allow users to "find common ground in their beliefs and interests" [49] and provide "opportunities to interact with people who share similar interests" [44]. According to Horrigan [50] the most popular online groups are professional groups, groups for people who share a hobby, an interest or a lifestyle, fan groups of sports teams or TV shows, local community groups and health-related support groups. Hence, most online groups are some sort of interactive information media used primarily for satisfying specific cultural interests or practical needs of participants. However, the existing empirical research yields mixed results. Lee et al [44] have established that self-reported *frequency of group feature use* was unrelated to social capital. Norris [51], having used Pew Internet & American Life project survey data, found that reported membership in certain *types of SNS groups* contributed to bridging and bonding social capital more than membership in others, although all contributions were modest. Finally, Lee and Lee [49] showed that the use of online groups is associated with perceived outcomes of social capital. Thus, the impact of online group membership on social capital remains under-researched. Given this, we assume that extensiveness of group membership should positively affect network brokerage, because it can provide access to more non-redundant contacts.

**H3:** The number of online groups a user belongs to is positively related to a user's brokerage capacity.

Finally, as we study a social network of a geographically localized population, there is a need to test how user's adherence to and boundedness by a local network might affect his/her social capital. Since social media unable to overcome a cognitive constraint of the size of a personal social network [52] we assume that a larger fraction of local ties (and, therefore, fewer external ties) among a user's SNS friends should positively relate to within-city social capital. However, this hypothesis is context-sensitive and less applicable to international cities with

intensive migration, where social ties outreaching other places will be more prevalent and are likely to be an indicator of rich social capital. Thus, this hypothesis is limited to within-city social capital and cannot be generalized to the level of general social capital that includes any social ties.

**H4:** Share of local friends among all user's friends is positively related to within-city social capital.

Thus, the available research suggests that there are three main types of online user behavior based on main SNS functions that can contribute to accumulation of social capital: sharing identity information in a user profile, communicating via features available on individual pages and participating in online groups. Building upon these findings, in this research we seek to test how the use of these SNS features is related to social capital in a location-bounded network.

## Data and methods

This we measures and examines social capital using the online friendship graph of an entire geographically localized population from a medium-sized city. Data was obtained from the largest Russian-speaking SNS, VK (also known as VKontakte, http://vk.com) [53], and we focused on the Russian city of Vologda. This city was selected because it is a typical medium-sized Russian city (population 313,012) with an average standard of living (38 out of 85 Russian regions by GRP) [54] and level of Internet penetration [55]. We avoided cities with specific ethnic composition, as well as cities close to the Russian borders, Moscow and St. Peterburg because they tend to have specific migration patterns. While this does not liberate our research from the limitations of a case study approach, the results obtained from this are more representable of others across of Russia rather than using an outlier. Although more research is needed to reveal which Vologda patterns are universal, and which are unique.

### Dataset: Vologda friendship network and online user behavior

VK provides functionality similar to Facebook. The data was collected automatically using an official VK application programming interface (API). The dataset includes all within-city friend links and information from users' profiles, such as counts of communication activity from their pages and metadata (gender, age, interests, education, etc). A separate subset is the data on features of VK groups to which users belong (See Table 2 for full list of measures). The datasets used in this study are available from the Open Science Framework: https://osf.io/hw2b6/.

At all stages, we only used open data, legally available from the VK server—that is data that can neither be hidden, according to the VK terms, nor the data a user chooses not to protect with privacy settings. Data was anonymized after the downloading. The research protocol was approved by the Institutional Review Board of the National Research University Higher School of Economics.

According to our research of VK random samples, city of residence is usually available for two thirds of non-dormant accounts, while friend lists could not be hidden at the time of data collection, which makes our data fairly complete. Most data we used was fully available, or variables were constructed so avoid missing data. More details regarding the completeness of data is given in Table 2.

Our initial population was 286,994 users who declared Vologda as their city of residence as of the date of data collection (04.09.2017). After filtering out banned users and those whose last visit to the VK was earlier than 01.06.2016, we constructed the graph of reciprocal

friendship ties that included 196,684 users connected by 9,800,107 edges (graph metrics are shown in Table 1). After additional filtering, the final sample comprised of 194,601 users who constituted the giant connected component used for regression analysis.

The descriptive analysis of the Vologda VK network shows that its structural characteristics (see Table 1) are similar to those of other online social networks [56] and certain random graph models. It is particularly similar to Watts-Strogatz small-world network model in terms of transitivity and modularity computed with Louvain community detection algorithm. At the same time, our network is similar to Barabasi-Albert scale-free model in terms of degree centralization. Thus, we can say that this network consists of internally dense clusters and star-type nodes with a very high centrality, which is in line with the vision of a city as a network of networks [24, 57]. Vologda VK network structurally is also similar to another VK friendship network from the city of Izhevsk [58], in particular by transitivity, assortativity by degree and modularity.

## Measures

**Social capital.** As mentioned above, in this study we follow a structural, or network conceptualization of social capital. SNS friendship is a relationship based on mutual recognition that makes friend's updates and posts visible in a user's newsfeed [59]. The latter is important for receiving social news, maintaining relationships and for responding to help requests [19, 60]. In this research we use both local metrics based on immediate user ties and global metrics based on ties beyond users' ego-networks. For closure, which by its nature can only be local, we use transitivity (local clustering coefficient) [61] calculated as the share of closed triads among all the triads in an ego-network. It reflects the embeddedness of an individual in a tightly connected group. For brokerage we use betweenness centrality [62], a global metric calculating the number of the shortest paths passing through a node. It estimates an individual's ability to bridge disconnected and distant nodes or clusters at the scale of an entire network. Finally, we use eigenvector centrality [63] accounting for degree of connected nodes as a global metric capturing Lin's idea about actor's social capital dependence on status, resources or, in our case, social ties of others related to them. The list of measures is given in Table 2.

**Table 1. Graph metrics for Vologda friendship network and random graph models.**

| Metrics | VK graphs | | Random graph models | | |
|---|---|---|---|---|---|
| | Vologda (giant component) | Izhevsk | Erdos-Renyi | Scale-free | Small World (p = 0.3) |
| Nodes | 196,630 | 477,057 | 196,630 | 196,630 | 196,630 |
| Edges | 9,800,077 | 17,742,662 | 9,800,077 | 9,830,225 | 9,831,500 |
| Density | 0.000507 | 0.000155 | 0.000507 | 0.000508 | 0.000508 |
| Average degree | 99.680 | 74.384 | 99.680 | 100 | 99.987 |
| Connected components | 1 | | 1 | 1 | 1 |
| Diameter | 9 | | 4 | 4 | 4 |
| Average geodesic distance | 3.15546 | 3.590 | 2.957603 | 2.889812 | 2.998528 |
| Transitivity (global clustering coefficient) | 0.080921 | 0.090 | 0.000508 | 0.003621 | 0.087468 |
| Average clustering coefficient (Watts-Strogatz) | 0.130105 | | 0.000508 | 0.003529 | 0.088209 |
| Average aggregate constraint | 0.065472 | | 0.010144 | 0.013402 | 0.011962 |
| Centralization degree | 0.033852 | | 0.000245 | 0.022046 | 0.000168 |
| Centralization betweenness | 0.011070 | | 0.000012 | 0.006248 | 0.000009 |
| Assortativity by degree | 0.140230 | 0.162 | 0.000289 | 0.003023 | 0.000017 |
| Modularity | 0.362820 | 0.377 | 0.070148 | 0.084263 | 0.361638 |
| Clusters | 21 | | 8 | 9 | 4 |

**Table 2. Study variables.**

| Variable | Description |
|---|---|
| *Dependent Variables** | |
| Transitivity (local clustering coefficient) | Ratio of all existing ties between alters in an ego-network to all possible ties between alters in this ego-network. Varies between 0 and 1, where 1 is the fully connected ego-network [61]. Indicator of network closure. |
| Betweenness centrality | Number of shortest paths going through the vertex [62]. Indicator of brokerage capacity. |
| Eigenvector centrality | Relative score of a node's centrality that depends on centralities of the node's neighbors [63]. Indicator of global centrality. |
| *Independent Variables* | |
| ***Control variables*** | |
| Age | User age indicated in the profile (100% available with the used API) |
| Gender | User gender indicated in the profile (100% available with the used API) |
| Occupation type | Availability of the main occupational activity (school, university, work, none) |
| Duration | Number of days since the date of a user's registration in VK (100% available with the used API) |
| ***Availability of identity information*** | |
| Photos | Total number of photos publicly shared on a user's page |
| Audios | Total number of audio records publicly shared on a user's page |
| Interests & beliefs | Number of fields filled in a user's profile and available publicly; they reflect interests, beliefs and values: «Attitude to alcohol», «Attitude to smoking», «Religion/World view», «Personal priority/the main thing in a life», «Important in others», «Political views», «Inspired by», «Activity», «About me», «Interests», «Favorite music», «Favorite movies», «Favorite TV shows», «Favorite games», «Favorite books», «Favorite quotes». Varies between 0 and 16. |
| School | Public availability of information about user's school on the page (0 or 1) |
| University | Public availability of information about a user's university on the page (0 or 1) |
| Relatives | Public availability of links to pages indicated as relatives on a user's page (0 or 1) |
| ***Communication activity**** | |
| User's posts | Number of posts made by a user on his/her wall |
| Others' posts | Number of posts made by other users on a user's wall |
| Likes | Total number of likes to posts on a user's wall (regardless of authorship) |
| Comments | Total number of comments to posts on a user's wall (regardless of authorship) |
| Reposts | Total number of reposts of posts from a user's wall (regardless of authorship) |
| ***Multiple groups membership**** | |
| Online groups | Number of online groups in VK in which a user is a member |
| ***Users' adherence to within-city network*** | |
| Share of local friends | Share of user's fiends residing in Vologda among all user's friends in VK (available for all users in the sample based on approx. two thirds of their friends) |

*VK allowed for no more than five hidden friends who usually could be retrieved from the pages of their counterparts. Completeness of this data is close to 100%.

**These data are incomplete which is why three strategies of dealing with the missing data were applied (including modeling only those observations for which full data was available). As all models produced very similar results, we report the most complete models where missing observations were coded as zeros, and all observations were kept in the model.

**Availability of identity information.** This category includes all fields from the users' profiles that were reasonably well populated. As we were interested in the amount, not in its content, of publicly available identity information, we used simple counts for such variables as Photos, as well as the additive index of Interests and Beliefs. If the data was not shared publicly by a user, this was coded as zero.

**Communication activity.** Outgoing communication activity has been measured with only one variable–the number of posts made by a user on this/her wall. Incoming communication has been measured by a range of simple metrics including the absolute number of likes, comments and reposts on a user's wall (regardless of authorship, but with the prior knowledge that they are mostly not authored by the wall owner), and the number of posts made by other users on a user's wall. Later, reposts were excluded from the final analysis due to multicollinearity. An aggregate index of activity dropped out from the final models because it had a smaller explanatory power than the variables from which it had been constructed.

**Multiple group membership** has been measured with only one variable–the number of online groups to which a user belongs.

**Users' adherence to within-city network** was measured as the share of friends from Vologda among all friends of a user.

## Data analysis

R (version 3.5.1) was used to execute all computations. Network metrics were computed using the 'igraph' R package. The natural log transformation was performed for all dependent variables and for a number of independent variables to correct for the skewedness in the data. Multiple linear ordinary least squares regression was used ('lm' function in R), despite its limitations for clustered data, as inference for network predictions stays one of the unresolved problems in the field [64]. The results of the statistical analysis do not necessarily imply a causation between variables. The R code for data transformation and regression analysis is available as a supplementary file S1 File.

## Results

Table 3 presents the final regression models with betweenness centrality, transitivity and eigenvector centrality in the social network of Vologda as dependent variables. The higher the betweenness centrality, the more structural holes and bridging ties are around a user, which may be used to gain brokerage benefits. The higher the transitivity, the more likely the formation of closed triangles among user's neighbors and the higher the density of connections among them. The higher the eigenvector centrality, the higher the aggregate centrality of user's friends. Brokerage regression model (betweenness centrality) demonstrates quite high explanatory power with 49% of explained variance (adjusted $R^2$ = 0.487). The model for network closure (transitivity) demonstrates moderate explanatory power and explains 33% of the variance (adjusted $R^2$ = 0.326). Finally, the model for eigenvector centrality explains 40% of the variance (adjusted $R^2$ = 0.407). Overall, regression models demonstrate explanatory power comparable to or a little higher than obtained in the existing research [9, 20, 44, 45, 60].

It is important to note that firstly, nearly all effects are significant, however we should keep in mind that with our sample size more attention should be paid to the effect size than to its significance. Most variables have small regression coefficients and tend to randomly flip their signs when model parameters are slightly changed. This means that these independent variables have no stable relation to the dependent variables. However, six variables highlighted in Italic have demonstrated the strong and stable pattern of association across all models. Models based on only those six variables explain 92–95% of the variance explained by the full models.

Secondly, closure has consistently demonstrated the inverse direction of association with most independent variables, as compared to the two other types of social capital. All three dependent variables turned out to be highly correlated, especially when logarithmized, with transitivity being negatively related to the other two. This indicates the existence of a trade-off between closure and brokerage acknowledged by Burt [23], however, it contradicts his

**Table 3. Multiple linear regression showing association of structural social capital with online user behaviors.**

| Variable | Brokerage | | Closure | | Global centrality | |
|---|---|---|---|---|---|---|
| | Betweenness centrality | | Transitivity | | Eigenvector centrality | |
| | Beta (95% CI) | *P*-value | Beta (95% CI) | *P*-value | Beta (95% CI) | *P*-value |
| **Control variables** | | | | | | |
| Gender (male) | 0.030 (0.014, 0.047) | <.001 | 0.063 (0.057, 0.069) | <.001 | -0.067 (-0.084, -0.051) | <.001 |
| Age | -0.092 (-0.092, -0.091) | <.001 | -0.015 (-0.016, -0.015) | <.001 | -0.006 (-0.007, -0.006) | <.001 |
| Occupation: school | 0.004 (-0.036, 0.044) | .835 | 0.053 (0.039, 0.067) | <.001 | -0.102 (-0.142, -0.062) | <.001 |
| Occupation: university | 0.031 (0.009, 0.053) | .006 | -0.040 (-0.048, -0.033) | <.001 | 0.062 (0.040, 0.084) | <.001 |
| Occupation: work | 0.079 (0.056, 0.102) | <.001 | -0.045 (-0.053, -0.037) | <.001 | 0.089 (0.066, 0.112) | <.001 |
| *Duration* | 0.214 (0.214, 0.214) | <.001 | -0.222 (-0.222, -0.222) | <.001 | 0.214 (0.214, 0.214) | <.001 |
| **Identity & personality information** | | | | | | |
| *Photos* [a] | 0.168 (0.162, 0.174) | <.001 | -0.117 (-0.119, -0.115) | <.001 | 0.126 (0.120, 0.132) | <.001 |
| Audios [a] | -0.008 (-0.012, -0.005) | <.001 | -0.019 (-0.020, -0.018) | <.001 | -0.002 (-0.006, 0.001) | .260 |
| Interests & believes [a] | 0.0002 (-0.013, 0.014) | .974 | -0.015 (-0.020, -0.010) | <.001 | 0.050 (0.037, 0.064) | <.001 |
| School | -0.020 (-0.045, 0.005) | .121 | 0.019 (0.010, 0.028) | <.001 | -0.020 (-0.045, 0.005) | .116 |
| University | -0.014 (-0.045, 0.017) | .389 | -0.004 (-0.015, 0.007) | .486 | 0.021 (-0.011, 0.052) | .198 |
| Relatives | 0.007 (-0.017, 0.031) | .555 | 0.034 (0.026, 0.043) | <.001 | -0.055 (-0.079, -0.030) | <.001 |
| **Communication activity** | | | | | | |
| *User's posts* [a] | -0.171 (-0.178, -0.164) | <.001 | 0.146 (0.144, 0.149) | <.001 | -0.023 (-0.030, -0.016) | <.001 |
| Others' posts [a] | -0.020 (-0.025, -0.015) | <.001 | 0.083 (0.081, 0.085) | <.001 | -0.042 (-0.047, -0.037) | <.001 |
| *Likes* [a] | 0.380 (0.373, 0.387) | <.001 | -0.320 (-0.322, -0.317) | <.001 | 0.206 (0.199, 0.212) | <.001 |
| Comments [a] | 0.023 (0.016, 0.030) | <.001 | -0.019 (-0.021, -0.017) | <.001 | 0.007 (-0.0004, 0.014) | .066 |
| **Multiple group membership** | | | | | | |
| *Online groups* [a] | 0.246 (0.240, 0.252) | <.001 | -0.183 (-0.185, -0.180) | <.001 | 0.234 (0.228, 0.241) | <.001 |
| **Users' adherence to within-city network** | | | | | | |
| *Share of local friends* | 0.285 (0.241, 0.329) | <.001 | -0.157 (-0.173, -0.141) | <.001 | 0.179 (0.137, 0.221) | <.001 |
| Constant | 0.000 (-0.055, 0.055) | 1.0 | 0.000 (-0.020, 0.020) | 1.0 | 0.000 (-0.054, 0.054) | 1.0 |
| Observations | 186,962 | | 183,818 | | 191,772 | |
| Adjusted R [a] | 0.488 | | 0.325 | | 0.406 | |

Standardized beta coefficients, 95% confidence intervals (in brackets) and P-values are reported. Italicized variables demonstrated the strong and stable pattern of association across all models.

[a] log transformation.

argument regarding the complementary character of those two that should be possible in parallel with this trade-off. The most plausible explanation of this effect is as follows. High closure values are only possible in small networks which is confirmed by the strong negative correlation between closure and degree (number of friends). Once a user starts growing his/her network and especially accumulating bridging ties, the overall transitivity decreases, as the possible presence of a dense core is no longer captured by this metric.

As regression models for three types of social capital are similar, the results are reviewed according to independent variables further below.

## Control variables

Of all controlling variables only two have a stable effect on social capital. The first is usage duration–the time that passed since a user registered on VK. This result demonstrates the effect of preferential attachment mechanism on network formation–users who have been on VK for a longer period of time get an advantage in making additional ties which contributes to

their network brokerage and global centrality [65]. At the same time the association between duration and transitivity is negative, and this means that the longer an individual uses VK, the less closed his/her friendship network is. This, again, happens mostly because user networks grow with time and are therefore unable to preserve high values of transitivity. The second meaningful relation of social capital is to occupational status: those individuals who indicate work as their current occupation tend to have higher brokerage and global centrality, and lower closure, than those who do not declare or indicate other occupational status. The relation of other two types of occupation–secondary school and university studentships–to social capital is unstable across models, as is the relation of gender and age.

## Identity information

The overall contribution of identity information into social capital is fairly modest. The relatively large and stable effect has been demonstrated only by the number of photos which is positively related to betweenness and eigenvector centralities, and negatively–to transitivity. The larger the number of photos, the higher the network brokerage and global centrality, and the lower the network closure. The fact that it is photos that have an effect on social capital might have a number of explanations. First, photos are the most heavily used feature among all identity information features. Second, photos are what display users' identity by picturing events, objects and people a user finds to be important and worthy of displaying; hence, this feature facilitates finding a common ground between users. Thus, we partially confirm hypothesis H1.

## Communication activity

Outgoing communication activity measured as the number of user's posts on his/her wall was strongly negatively related to betweenness centrality, and positively–to transitivity. Results support the hypothesis H2a and indicate that outgoing communication affects user's existing friends rather than reaches a new audience, which ultimately leads to the growth of network closure.

Of all types of incoming communication activity, only the number of likes has a strong and stable effect on social capital: it is positively related to betweenness and eigenvector centrality, and negatively–to transitivity. Hence the more likes a user receives, the higher is his/her brokerage and global centrality in the location-bounded network. However, network closure decreases with the growth of the number of likes although one might expect that cohesive groups with tighter relations might produce more likes. Here, it is important to note that the direction of causality between likes and structural social capital may be inverse to what was initially assumed in our regression models. Likes can be an outcome of high popularity and good connectedness of a person on SNS. A surprising result is that the number of comments is weakly associated with brokerage and network closure in VK, which contradicts our assumption. Among other things, we expected that high a frequency of communication of others on a user's wall would increase mutual visibility of user's friends and the likelihood of friendship among them [19], which was to contribute to a higher transitivity. According to McLaughlin & Vitak [66], direct incoming communication was also to be related to bridging social capital which is similar to brokerage. Therefore, hypothesis H2b is partially supported, since not all types of engagement of incoming communication are found to be related to social capital.

## Multiple online group membership

The number of online groups in which a user is a member has a strong positive effect on brokerage and global centrality, and a strong negative effect on closure. These results clearly

support H3. Although most online groups from our study population do not relate to Vologda and even locally oriented online groups are usually open for any users regardless of their location, belonging to a larger number of groups strengthens social capital in a network of geographically proximate ties. This paradox might occur because, although group members have a greater chance of meeting people from other cities, their chance to meet and befriend a person from their own town is still higher than if they were to search for friends randomly outside of online groups.

## Users' adherence to a within-city network

The share of friends located in Vologda among all user's VK friends is normally distributed–this means that the majority of people tend to have relatively even proportions of friends within and outside the city, while only minorities are embedded entirely either within or outside Vologda. The share of local friends has a positive effect on brokerage and global centrality and a negative effect on closure. Hence the more adherent a user is to the city of his/her residence, the higher his/her brokerage and global centrality is in the within-city friendship network. Since social media is unable to extend the size of a personal social network beyond cognitive limits [52], and the entire social network is quite clustered, therefore, local friends of a user, with a high share of them among all his/her VK friends, are more likely to be distributed across different clusters than for someone with lower share of local friends. Thus, H4 is supported.

## Discussion

### Transitivity as problematic indicator of network closure

Burt [23, p. 225] argues that closure and brokerage are complementary network structures augmenting each other in creating social capital. The maximum individual advantage is achieved at extreme levels of both brokerage and closure, when an actor simultaneously belongs to a cohesive group and has bridging ties beyond it. However, since our data indicates transitivity (as an indicator of closure) is inversely related to betweenness (the Spearman correlation is -0.54), empirically their relationship turns out to be rather mutually exclusive than complementary. This finding partially coincides with Brooks et al. [20] who found that transitivity in friendship ego-networks negatively correlated with the number of clusters and modularity (which are indicators of network brokerage). Thus, a drawback of transitivity is that it actually measures the overall tendency of an ego-network to form a single clique but not the cliquishness of some or all clusters in an ego-network. Transitivity might be equally low for same-size ego-networks with very different structures: both for those with cohesive but disconnected clusters (i.e. with high closure by Burt's definition), and for those with looser but more interconnected clusters (with low closure). Burt stressed that closure is a feature of a group/cluster, and since an individual can engage with a number of distinct clusters, another metric is needed to capture how dense separate clusters in a user's network are. In our research, we see that the entire city-bounded network is a loose collection of tighter clusters, and transitivity drops rapidly for those engaged with more than one cluster. Such engagement should not exclude high closure, but transitivity does not account for it. This means that transitivity is not good enough as an indicator of network closure.

### Online groups as a source of network brokerage

We have found out that the more online groups a user belongs to, the higher his/her network brokerage is, i.e. the more various social milieus a user connects to and bridges between. In a

large and heterogeneous social network bounded within the same city, membership in online groups, many of which are not associated with the city, paradoxically contributes to the gain of geographically proximate bridging ties. A possible mechanism causing this effect warrants further discussion. Formally, being a member of an online group and forming friendships with its members are two distinct types of online behavior. However, there is a substantial body of literature exploring network structures of different types of online groups including online forums [67], social news sites [68], twitter #hashtag communities [69, 70], Facebook groups [71], and VK groups [72–74]. These studies demonstrate that although these platforms have different network patterns [75], dense and tightly connected clusters of friendship are usually formed in most online groups. This suggests that even a single friendship with another group member may provide access to a whole bunch of social contacts, and a user joining such clusters in multiple groups inevitably becomes a broker. Thus, the more online groups a user joins in SNS, the higher the chance of having more non-redundant local connections.

## Disclosed identity information and social lubricant effect

Social lubricant effect appears when identity information in SNS is used for searching and establishing common ground between users [22, 60]. While previous research [38] found that the amount of identity information has a weak positive relation to the number of friends on Facebook, we find the effect of most types of such information so small that it is not able to substantially affect social capital. This result is consistent with an argument of Lin [76] who claims that adopting more complex measures of users' online behavior is a more fruitful approach for an analysis of personal social outcomes. This is because a user does not display a single behavior online but rather embodies an integrated social "grooming" style. Thus, further nuanced research is required to investigate whether comparable identity information, such as the same school or common interests, really increases the probability of friendship tie formation more than the mere amount of information. Meanwhile, the number of photos increases the network brokerage, regardless of their content. Among all other types of identity information, a photo is the most emotional and easy-to-consume way of self-disclosure. Posts with photos are known to generate far more likes than regular posts [77], while some research finds that positive feedback (of which likes are an example) is positively related to perceived bridging social capital and even mediates the effect of self-disclosure [78]. Therefore, compared to profiles with relevant, but non-visualized information, profiles photos rich are more likely to quickly provide information sufficient for establishing common ground with a social information seeker and to attract positive feedback from "well-matching" seekers. This might be a possible explanation of why specifically photos play the role of social lubricant on SNS.

## Engagement of other users as an attention signaling activity

The fact that engagement of others in the form of likes contributes to brokerage, but not to closure, deserves special consideration. If explained by relationship maintenance behavior, engagement of others on a user's wall should increase brokerage of others, not of the wall owner. Those who use the friend's wall become exposed to friends of the wall owner, and therefore can establish new ties possibly including non-redundant contacts. In this case, brokerage of the wall owner should decrease, while closure should increase, which is exactly the opposite of our findings. Unlike other forms of engagement (posts and comments), likes have less ability to cause addressed reaction from others because authors of likes are less visible to others and less distinct from each other. Therefore, likes can hardly contribute to network closure of a wall owner's. At the same time, Burke et al. [17] who also found that incoming (and not outgoing) communication is positively related to bridging capital, offer the following

explanation: it is the feedback that signals a user about the existence of a tie. Further developing this claim, we may say that outgoing communication, i.e. broadcasting on the user's wall, is only an attempted relationship maintenance activity. The reciprocal act of communication is a confirmation of this activity being successful. It is likes–the low-cost signals of attention and social approval–that allow such confirmation [21]. Given our earlier reflections on the direction of causality between likes and social capital, we can assume that high numbers of likes plausibly present confirmation of the gained brokerage ability rather than its cause.

## Conclusion

This study is, to the best of our knowledge, the first examination of the effects of SNS user behaviors on online social capital within a large geographically localized population–in this case, a medium-sized city. As opposed to studies of independent ego-networks typical to the field, the focus on a city has first allowed us to examine social capital calculated from an entire network. This, in turn, has allowed us to account for the effect of indirect connections–those leading to the "right" person [24]–and the effect of social proximity to the network hubs–that is, possession of ties leading to influential persons. Second, our approach has given us an opportunity to examine geographically proximate relationships whose advantage over other user's online ties is that they allow access to potentially more tangible and location-related resources such as information regarding local jobs [31], housing rentals, medical aid [32] or childcare services [33].

We found that the global structure of the location-bounded network presents a combination of small-world and core-periphery graphs containing dense clusters and star-type nodes with outlying centralities. This suggests the presence of a hierarchical structure in the network. Although this relatively large community breaks into small sub-communities (high global transitivity), it is also connected by a small number of city-level hubs (as indicated by high degree and betweenness centralization, and comparatively high assortativity by degree). Further, the city-level network has no clear boundaries since the majority of users have equal proportions of their friends inside and outside the city of their residence. However, the adherence to and isolation within the city network is directly related to users' within-city social capital, especially to within-city brokerage. The availability of rich geographically related network data on VK provides great potential for further comparative analysis of regions, cities, or urban and rural communities, and thus provides a means of overcoming the limitations of a case-study approach.

The focus on an entire geographically localized network has made possible our major finding regarding the effect of multiple online group membership on within-city social capital and its interpretation. Surprisingly, this obvious hypothesis had not been tested before, perhaps, due to difficulty in obtaining data. We revealed that globally measured social capital, including brokerage, is positively related to the number of groups a user belongs to, while closure demonstrates an inverse relation. Online groups naturally serve as gateways to new social milieus where new friends may be acquired, for whom a user becomes a broker, connecting them to the rest of his/her network. Most plausibly, it is online communities–being smaller, more interactive and thus more suitable for practical needs–that play a leading role here, while pages function more as mass media. Paradoxically, social capital gain in a within-city network is associated with multiple memberships in online groups although most of them have no location or are located outside the studied city. Perhaps, the effect of groups on social capital might be stronger if local groups could be singled out from all groups for each user, or if social capital was calculated based on all ties, including location-independent friendships. These are all questions for potential further research.

In this paper we have also shown that certain types of outgoing (photos) and incoming (likes) activities in a users' profile are positively related to his/her brokerage and global centrality in a location-bounded network. While photos display user's identity and thus provide social information seekers with necessary context for linking with the page owner, likes appear to work differently. They signal page owners that their ties are "alive" and usable and may serve rather as consequences (or indicators) of high global centrality and brokerage than as antecedents. A limitation of our study is that we did not use data about a user's activity outside of their walls, such as liking or commenting on a friends' page, which is an important part of social grooming behavior. This is one way to further develop this research.

Finally, we found that transitivity strongly and negatively correlates with betweenness centrality. This means that transitivity is hardly a good measure for closure, because closure should rather complement brokerage than replace it. Combined with findings of Brooks et al [20], this calls for deeper investigation into the empirical and conceptual validity of network measures to social capital concepts. Ultimately, it calls for further clarification of the concept of social capital.

## Supporting information

**S1 File. R code for data transformation and analysis.**
(R)

## Author Contributions

**Conceptualization:** Yuri Rykov, Olessia Koltsova.

**Data curation:** Yuri Rykov, Yadviga Sinyavskaya.

**Formal analysis:** Yuri Rykov, Olessia Koltsova.

**Funding acquisition:** Olessia Koltsova.

**Investigation:** Yuri Rykov.

**Methodology:** Yuri Rykov.

**Project administration:** Yadviga Sinyavskaya.

**Writing – original draft:** Yuri Rykov.

**Writing – review & editing:** Yuri Rykov, Olessia Koltsova, Yadviga Sinyavskaya.

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
