## [Decision Letter · Decision Letter 0]

29 Jan 2020

PONE-D-19-35848

Effects of user behaviors on accumulation of social capital in an online social network

PLOS ONE

Dear Dr Rykov,

Thank you for submitting your manuscript to PLOS ONE. After careful consideration, we feel that it has merit but does not fully meet PLOS ONE’s publication criteria as it currently stands. Therefore, we invite you to submit a revised version of the manuscript that addresses the points raised during the review process.

Although reviewer 1 recommend to accept the manuscript after improving its clarity, Reviewer 2 raised several concerns that we ask you to consider in the revision of your manuscript.

We would appreciate receiving your revised manuscript by Mar 14 2020 11:59PM. To enhance the reproducibility of your results, we recommend that if applicable you deposit your laboratory protocols in protocols.io, where a protocol can be assigned its own identifier (DOI) such that it can be cited independently in the future. For instructions see: http://journals.plos.org/plosone/s/submission-guidelines#loc-laboratory-protocols

We look forward to receiving your revised manuscript.

Kind regards,

Alexandre Bovet, Ph.D.

Academic Editor

PLOS ONE

Reviewers' comments:

Reviewer's Responses to Questions

**Comments to the Author**

1. Is the manuscript technically sound, and do the data support the conclusions?

Reviewer #1: Yes

Reviewer #2: Yes

2. Has the statistical analysis been performed appropriately and rigorously? 

Reviewer #1: Yes

Reviewer #2: Yes

3. Have the authors made all data underlying the findings in their manuscript fully available?

Reviewer #1: Yes

Reviewer #2: No

4. Is the manuscript presented in an intelligible fashion and written in standard English?

Reviewer #1: Yes

Reviewer #2: Yes

5. Review Comments to the Author

Reviewer #1: This is an excellent MS, and in the spirit of PLoS One philosophy I will not offer any criticisms other than to suggest that the English could be improved a bit in the second half (there are a few places where wording slips a bit and is incorrect). The positive comments I would have relate to the fact that these are data from a Russian SNS, so make an important contribution in their own right from the usual sources like Facebook. The results make some valuable new contributions to our understanding of how the online world works.

Reviewer #2: This study examined an entire city-bounded friendship network on VK. The topic is of research significance.

The theoretical framework (social capital) is clear. Methodologically, SNA is an ideal approach for the research topic. While my major concern for recommending publication is related to the reasoning of hypotheses and unclear arguments behind.

The H1 is a very interesting hypothesis. But more socio-psychological justification of the positive relationship between willingness of publicly display information and richness of social capital is needed. More literature review need to be done, for example, the privacy concern might prevent a person with rich social capital (at lease closure) release too much personal information on SNS.

The proposition of H2 is not persuasive. Associating SNS engagement with social capital is acceptable. But the author have not yet highlighted the logical inference why we can use such behaviour indicators to predict user’s social capital richness. There are many factors contribute to the SNS engagement intensity as well as a user’s social capital. Thus, it reads problematic to simply link up the two together. Actually the issue has been reflected by empirical results. Even though data results have showed that the number of likes was strongly correlated with the structural positions of a VK user, the correlation directions were different between closure and brokerage, and the other two engagement indicators found insignificant. The author may want to pay more attention to the symbolic meaning of Like in the SNS network. Giving Like is an impulsive but ambiguous action on the online social network, the quantity of Likes initiated by complex motivations might reflect the role of an SNS user’s engagement in the online network, but not necessary represent his or her social capital richness.

H4 is debatable but arguable. The hypothesis should be context-sensitive but not applicable to international city. Like international cities of Singapore and Hong Kong, divers social ties outreaching other places can be an indicator of rich social capital. Similar idea (i.e. ethnic composition) has been used to justify the selection of city in method section. I think this point can be mentioned during literature/theory discussion.

6. PLOS authors have the option to publish the peer review history of their article (what does this mean?). If published, this will include your full peer review and any attached files.

Reviewer #1: Yes: Robin Dunbar

Reviewer #2: No

---

## [Author Response · Author response to Decision Letter 0]

11 Mar 2020

Dear colleagues,

We very much appreciate the opportunity to revise our manuscript: “Effects of user behaviors on accumulation of social capital in an online social network” - Submission PONE-D-19-35848.

We would like to thank the editorial team and the reviewers for their engagement with our manuscript, fair comments and helpful suggestions for improvement. We have revised and changed the manuscript accordingly and present a point-by-point response to reviewers’ comments further below. 

We hope the revised manuscript meets your expectations and would be happy to make any further revisions or improvements if needed. 

Best regards

Authors

RESPONSE TO REVIEWERS

Reviewer #1

This is an excellent MS, and in the spirit of PLoS One philosophy I will not offer any criticisms other than to suggest that the English could be improved a bit in the second half (there are a few places where wording slips a bit and is incorrect). The positive comments I would have relate to the fact that these are data from a Russian SNS, so make an important contribution in their own right from the usual sources like Facebook. The results make some valuable new contributions to our understanding of how the online world works.

Response:

Thank you very much for such a positive feedback on our manuscript. Manuscript was proofread and edited, and English was improved throughout the manuscript as suggested.

Reviewer #2

Comment 1

The H1 is a very interesting hypothesis. But more socio-psychological justification of the positive relationship between willingness of publicly display information and richness of social capital is needed. More literature review need to be done, for example, the privacy concern might prevent a person with rich social capital (at lease closure) release too much personal information on SNS.

Response:

Thank you for your suggestion. The scope of the literature review was extended by two additional arguments for more profound justification of the H1. First, the positive role of online self-disclosure as a basis for converting the so-called latent ties (Haythornthwaite, 2005) into weak/strong ones by providing social context was discussed. This mechanism explains how the online self-disclosure may be converted into positive social capital outcomes. 

Second, the inhibiting effect of privacy concerns on online self-disclosure was addressed (Ellison, 2011; Hogan, 2010; Vitak, 2012). In Stutzman et al. (2012) the negative relationship between usage of strict privacy settings (i.e. non-disclosure) and bridging social capital were revealed as well as limited benefits of such behaviour on bonding social capital. These results provide the ground for our suggestion about the positive relationship between the extent of disclosed profile information and users' network social capital.

Comment 2

The proposition of H2 is not persuasive. Associating SNS engagement with social capital is acceptable. But the author have not yet highlighted the logical inference why we can use such behaviour indicators to predict user’s social capital richness. There are many factors contribute to the SNS engagement intensity as well as a user’s social capital. Thus, it reads problematic to simply link up the two together. Actually the issue has been reflected by empirical results. Even though data results have showed that the number of likes was strongly correlated with the structural positions of a VK user, the correlation directions were different between closure and brokerage, and the other two engagement indicators found insignificant. The author may want to pay more attention to the symbolic meaning of Like in the SNS network. Giving Like is an impulsive but ambiguous action on the online social network, the quantity of Likes initiated by complex motivations might reflect the role of an SNS user’s engagement in the online network, but not necessary represent his or her social capital richness.

Response:

Thank you for the comment. So far as we understand this comment on H2, it stems from the – now obvious – lack of clarity in our manuscript which made the reviewer confused with this hypothesis. To make this clear, we have introduced the following changes. (1) We splitted H2 into two sub-hypotheses: H2a regarding the role of outgoing communication activity, and H2b – regarding incoming communication activity. (2) As suggested by the reviewer, we revise the logical inference for H2a and H2b right before they appear in the text. (3) We remade regression analysis by replacing total number of posts and a share of others’ posts with a number of a user’s own posts and a number of posts made by others. Nevertheless, this did not lead to any significant changes in the results, so interpretations left almost the same. (4) Throughout the text, we now consistently refer either to outgoing communication activity (broadcasting) or to incoming communication (or, in other words, engagement of others on a user’s wall). We hope that revisions helped make the manuscript more consistent and conceptually clearer.

Comment 3

H4 is debatable but arguable. The hypothesis should be context-sensitive but not applicable to international city. Like international cities of Singapore and Hong Kong, divers social ties outreaching other places can be an indicator of rich social capital. Similar idea (i.e. ethnic composition) has been used to justify the selection of city in method section. I think this point can be mentioned during literature/theory discussion.

Response:

Thank you for the helpful suggestion. We agree with the comment and believe that social ties outreaching other places can be an indicator of rich social capital not only in international cities, but also in most other places. We extended the paragraph on H4 and added this point to the section with literature review and hypotheses. However, since we initially focused our analysis on within-city social network (and provided a detailed rationale for this), we also emphasized possible explanation why grater boundedness to a local community can result in richer network brokerage.

References

1. Haythornthwaite C. Social networks and Internet connectivity effects. Inf Com Soc. 2005;8(2):125-147. doi:10.1080/13691180500146185 

2. Ellison NB, Vitak J, Steinfield C, Gray R, Lampe C. Negotiating privacy concerns and social capital needs in a social media environment. In: Trepte S., Reinecke L, editors. Privacy online. Springer, Berlin, Heidelberg; 2011. pp. 19-32.

3. Hogan B. The presentation of self in the age of social media: Distinguishing performances and exhibitions online. Bull Sci Technol Soc. 2010;30(6):377-386. doi:10.1177/0270467610385893 

4. Vitak J. The impact of context collapse and privacy on social network site disclosures. J Broadcast Electron Media. 2012;56(4):451-470. doi:10.1080/08838151.2012.732140 

5. Stutzman F, Vitak J, Ellison NB, Gray R, Lampe C. Privacy in interaction: Exploring disclosure and social capital in Facebook. In: Sixth international AAAI conference on weblogs and social media; 2012 June 4–7; Dublin, Ireland. AAAI Press. Toronto, Ontario, Canada, p.330-337.

---

## [Decision Letter · Decision Letter 1]

2 Apr 2020

Effects of user behaviors on accumulation of social capital in an online social network

PONE-D-19-35848R1

Dear Dr. Rykov,

We are pleased to inform you that your manuscript has been judged scientifically suitable for publication and will be formally accepted for publication once it complies with all outstanding technical requirements.

With kind regards,

Alexandre Bovet, Ph.D.

Academic Editor

PLOS ONE

Additional Editor Comments:

Please make clear in the introduction and discussion that the results of the statistical analysis does not necessarily imply a causation between your variables.

In particular, replace the term "prediction" or "predicting" with terms that do not necessarily imply a causation such as "association" or "correlation" or "showing an association".

Reviewers' comments:

Reviewer's Responses to Questions

**Comments to the Author**

1. If the authors have adequately addressed your comments raised in a previous round of review and you feel that this manuscript is now acceptable for publication, you may indicate that here to bypass the “Comments to the Author” section, enter your conflict of interest statement in the “Confidential to Editor” section, and submit your "Accept" recommendation.

Reviewer #2: All comments have been addressed

2. Is the manuscript technically sound, and do the data support the conclusions?

Reviewer #2: Yes

3. Has the statistical analysis been performed appropriately and rigorously? 

Reviewer #2: Yes

4. Have the authors made all data underlying the findings in their manuscript fully available?

Reviewer #2: (No Response)

5. Is the manuscript presented in an intelligible fashion and written in standard English?

Reviewer #2: Yes

6. Review Comments to the Author

Reviewer #2: The manuscript has been better improved. And I agree with Reviewer #1 that the submission make a contribution to the SNS scholarship by providing insights from data sources that are apart from major platforms like Facebook.

The authors have addressed my comments and concerns in the previous round of review and you feel that this manuscript is now acceptable for publication. Congratulations!

7. PLOS authors have the option to publish the peer review history of their article (what does this mean?). If published, this will include your full peer review and any attached files.

Reviewer #2: Yes: ZHANG Yin Nick

---

## [Editor Report · Acceptance letter]

7 Apr 2020

PONE-D-19-35848R1 

Effects of user behaviors on accumulation of social capital in an online social network 

Dear Dr. Rykov:

I am pleased to inform you that your manuscript has been deemed suitable for publication in PLOS ONE. Congratulations! Your manuscript is now with our production department. 

With kind regards,

on behalf of

Dr. Alexandre Bovet 

Academic Editor

PLOS ONE